# Dual-Port Butterfly Slot Antenna for Biosensing Applications

**DOI:** 10.3390/s25164980

**Published:** 2025-08-12

**Authors:** Marija Milijic, Branka Jokanovic, Miodrag Tasic, Sinisa Jovanovic, Olga Boric-Lubecke, Victor Lubecke

**Affiliations:** 1The Faculty of Electronic Engineering, University of Nis, 18000 Nis, Serbia; 2The Institute of Physics, University of Belgrade, 11000 Belgrade, Serbia; brankaj@ipb.ac.rs; 3Academy of Engineering Sciences of Serbia, Kraljice Marije 16, 11000 Belgrade, Serbia; 4The School of Electrical Engineering, University of Belgrade, 11000 Belgrade, Serbia; tasic@etf.rs; 5IMTEL Komunikacije a.d., 11070 Belgrade, Serbia; siki@insimtel.com; 6The Department of Electrical and Computer Engineering, University of Hawaii at Manoa, Honolulu, HI 96822, USA; lubecke@hawaii.edu

**Keywords:** butterfly slot antenna, coplanar waveguide transmission line, double-port antenna, microstrip feed line

## Abstract

This paper presents the novel design of a printed, low-cost, dual-port, and dual-polarized slot antenna for microwave biomedical radars. The butterfly shape of the radiating element, with orthogonally positioned arms, enables simultaneous radiation of both vertically and horizontally polarized waves. The antenna is intended for full-duplex in-band applications using two mutually isolated antenna ports, with the CPW port on the same side of the substrate as the slot antenna and the microstrip port positioned orthogonally on the other side of the substrate. Those two ports can be used as transmit and receive ports in a radar transceiver, with a port isolation of 25 dB. Thanks to the bow-tie shape of the slots and an additional coupling region between the butterfly arms, there is more flexibility in simultaneous optimization of the resonant frequency and input impedance at both ports, avoiding the need for a complicated matching network that introduces the attenuation and increases antenna dimensions. The advantage of this design is demonstrated through the modeling of an eight-element dual-port linear array with an extremely simple feed network for high-gain biosensing applications. To validate the simulation results, prototypes of the proposed antenna were fabricated and tested. The measured operating band of the antennas spans from 2.35 GHz to 2.55 GHz, with reflection coefficients of less than—10 dB, a maximum gain of 8.5 dBi, and a front-to-back gain ratio that is greater than 15 dB, which is comparable with other published single dual-port slot antennas. This is the simplest proposed dual-port, dual-polarization antenna that enables straightforward scaling to other frequency bands.

## 1. Introduction

Slot antennas of various shapes are becoming attractive for many modern wireless communications applications that need low-cost and low-profile components that can be seamlessly integrated into microwave frontends. A great effort has been made to improve upon the slot antenna design and to face different challenges posed by their potential use in Internet of Things (IoT) [1], wireless body area networks (WBANs) [2], RFID [3], and sensor network applications [4]. Slot antennas of a simple shape [5,6,7] can achieve a wide impedance bandwidth, but with very low gain, which is typically less than 5 dBi. Gain enhancement usually requires a complex feed structure [8], resulting in demanding antenna design, or the use of a slot antenna array which increases the overall antenna size [4]. The use of two feed ports is very common for slot antennas in order to achieve both right-hand and left-hand circular polarization [6] or to provide simultaneous use of two operating modes [2]. Therefore, the isolation between ports must also be considered and optimized to enable desired antenna functionality. Furthermore, slot antennas are usually used in complex communication systems with sensitive receivers; thus the front-to-back (FBR) gain ratio should be considered too.

This paper presents a simple two-port butterfly-shaped slot antenna designed at 2.4 GHz. The antenna is intended for full-duplex in-band applications in Doppler biosensing radars [9]. In recent years significant advancement in biomedical radar technology has been focused on miniaturization, increased accuracy, and enhanced sensitivity [10,11]. The proposed antenna with two mutually isolated ports, which can be used for simultaneous transmit and receive in short-range radars, allows for considerable miniaturization of the radar hardware. Since the radiating slots that form the butterfly arms of the antenna are orthogonally positioned, this antenna is also a double-polarized antenna that exhibits almost identical radiation patterns in the E and H planes. By implementing two types of feed methods, it is possible to simultaneously receive both vertically and horizontally polarized waves at the different antenna ports, as is required for comprehensive polarimetric analysis of physiological Doppler radar signatures [12,13]. Although the antenna is designed for the 2.4 GHz band, it can be easily scaled to any other frequency band due to its simplicity.

In our previous work on a one-port antenna with butterfly-shaped slots intended for K band applications, the antenna was fed by single coplanar waveguide (CPW) [14] or microstrip line (MSL) [15], both positioned along the coupling region. In this novel design of a two-port antenna at 2.4 GHz, the microstrip line is placed perpendicularly both to the coupling region and to the CPW line, to increase isolation between antenna ports. Unlike previously published two-port crossed-slot antennas [16,17,18,19,20,21], this novel design consists of two identical butterfly arms that are coupled through the region defined by the width and gap of the CPW feed line. In addition to the butterfly slot length and width dimensions, the parameters of the coupling region and the extension of the feed lines create more flexibility in the antenna optimization to achieve the same resonant frequency and input impedance at both antenna ports. Therefore, the proposed antenna does not require the complicated feed network and matching circuit that is needed in the other two-port crossed-slot antenna for adjustment of the resonant frequency and impedance at both ports [16,17,18,19,20,21]. The proposed butterfly slot antenna is designed using only one substrate with two conductive layers and fed by very simple CPW and MS lines, avoiding redundant bulky feed networks that introduce attenuation and increase overall antenna size. This is the simplest proposed dual-port, dual-polarization antenna design on a single substrate. Due to the antenna simplicity, a unique dual-port linear array is also proposed featuring eight butterfly slots and two feeding structures implemented on the same substrate, making antenna array modeling and fabrication simple and cost-effective.

After a brief discussion on recent advancements in the design of dual-polarized antennas in Section 2, Section 3 presents design and optimization results for a double-port butterfly slot antenna fed by microstrip and CPW lines, with and without extensions for fine adjustment of the antenna resonant frequency and the input impedance. The design and characteristics of a linear array of eight double-port butterfly slot antennas is given in Section 4. Experimental results for the antenna prototype are presented in Section 5 to validate the proposed design. A summary conclusion is given in Section 6.

## 2. Related Works

The recent growth of wireless communication services requiring an antenna with polarization diversity has generated significant interest in dual-polarized antennas. In addition to the low-profile, high-gain, and broad bandwidth that is required for an antenna in most wireless applications, multipolarized antennas also require high isolation between the ports and low cross-polarization levels.

Dual-polarized antenna can be realized using simple planar configurations such as patch [21,22] or slot [23,24] antennas, with only one substrate and two conductive layers. The low-profile and compact size of such antennas [23,24] are suitable for applications where miniaturization is critical. However, the gain of these antennas is usually below 6 dBi, which is not sufficient in modern wireless communication. Also, their radiation patterns have omnidirectional or bidirectional properties causing a small front-to-back ratio.

Two identical radiating elements crossed at the right angle [16,17,18,19,20,21,22] are commonly applied to achieve dual polarizations. Moreover, the crossed slots or patch antenna can lead to a dual-polarized radiation pattern with an identical radiation pattern in the E and H planes by implementing the appropriate feeding structure. The dual-polarized antenna presented in [22] features different radiation patterns in the E and H planes. For the antenna intended for a comprehensive polarimetric study of physiological Doppler radar signatures [12,13], where a multipath fading problem can occur, it is important that the antenna can simultaneously receive both vertically and horizontally polarized waves at the different antenna ports from different directions. Therefore, the design of an antenna featuring identical radiation patterns in the E and H planes, especially its feeding structure, can be challenging.

The antennas proposed in [16,17,18,19,20] use multilayered structures with more substrates to achieve dual-polarized properties and identical radiation patterns in the E and H planes. Although the radiation elements of the antennas in [16,17,18,19,20] have a simple structure of two crossed identical shapes, its feeding structure, usually at both ports, is complex. The antenna presented in [16] has fork-shaped feeding structures printed on a different substrate than the radiating elements, and it requires parasitic strips to reduce cross-polarization. The balun feeding structure is frequently used in the design of a dual-port dual-polarized antenna [17]. However, its realization with a pair of split ring lines on two substrates makes the antenna assembly complex and challenging. Moreover, the pair of coupled meander lines has to be carefully modeled to achieve the desired antenna performance [17]. The antenna from [18] consists of interdigital structures between the square patch edges and the thin metallic strips loaded with metallic vias. The radiating elements of the antenna presented in [19] use two substrates to improve antenna radiation characteristics. Moreover an additional substrate is added to realize a power divider necessary for one port excitation and additional isolation from second port interference [20].

The antenna proposed in [21] uses only one substrate; however its feeding structure, implemented with two coaxial lines with inner and outer conductors excited out of phase, is complex.

The aim of the presented work is to design a dual-polarized antenna with an identical radiation pattern in the E and H planes for both ports, with a simple feeding method and a good front-to-back ratio. Two identical butterfly slots crossed at the right angle were considered with a reflector that suppresses back radiation. The butterfly slot antenna can be easily integrated into an array employing the CPW series and MSL corporative feeding network. The array can be easily realized by simultaneously printing radiating elements and series feeding on the top side and a corporate feeding network on the bottom side. The array does not need balun as in the array in [25], vias between the array elements and corporative feeding network as in the array in [26], or 50 Ω coaxial lines between the antenna elements and dividers as in the array in [27].

## 3. Butterfly Slot Antenna

The proposed butterfly slot antenna is positioned on the top side of the FR4 substrate featuring dielectric constant *ε*_r_ = 4.3 and thickness *h* = 1.58 mm. The flat reflector is placed at the distance *λ*_0_/4 from the slot antenna, where *λ*_0_ is the wavelength in vacuum at the frequency *f*c = 2.4 GHz. Figure 1 depicts a butterfly slot radiating element composed of two equal butterfly wings with the following dimensions: widths *a* and *a*_1_ and length *b*. It can be observed that each slot wing is formed by two bow-tie slots situated at the right angle [14,15]. There is the coupling region between slot wings with dimensions *w* and *g* corresponding to the width and gap of the CPW feeding line. The properly selected antenna feeding method can significantly improve the antenna’s performances, particularly its bandwidth. Both the CPW line [3,7] and the microstrip line [5,8] are the most commonly used feeding structures of single-fed slot antennas. The main problem in the design of a dual-port antenna is to achieve the same resonant frequency when the antenna is excited from different ports and also to adjust the input impedance at both ports to 50 Ohm, avoiding the additional matching circuits at the antenna inputs. The unique shape of the proposed butterfly slot antenna allows optimization of seven antenna parameters: slot length *b*, two slot widths *a* and *a*_1_, the dimensions of the coupling region *w* and *g*, and two lengths of the feeding line extensions to achieve that goal. This feature enables an efficient design with a single substrate that does not demand additional substrates with more conductive layers for matching circuits necessary to adjust the resonant frequency and impedance at both ports.

### A Double-Port Feeding of the Butterfly Slot Antenna

With the rapid growth of various wireless applications, the study and design of antennas with two or more ports have gradually attracted tremendous attention [2,16,17,18,19,20,21]. Figure 2 presents a dual-port butterfly slot antenna intended for full-duplex in-band applications with two feeding structures consisting of a CPW and microstrip line. Only two conductive layers are used to achieve requested impedance at both ports at the frequency *f*c.

Two versions of the butterfly slot antenna are considered: Antenna 1 is designed with a CPW line placed into the coupling region between two butterfly wings and a microstrip line situated orthogonally to the coupling region (Figure 2a), while Antenna 2 consists of a CPW line extended for length *l*_ce_ over the antenna center to fine-tune the antenna resonant frequency (Figure 2b). The engaged CPW feeding line has width *w* = 3.1 mm and gap *g* = 0.3 mm, giving the characteristic impedance *Z*_c_ = 50 Ω at the center frequency *f*_c_. At the end of the CPW line, with length *l*_c_ from the center of the slot, the SMA connector is placed.

The microstrip line having width *w*_m_ = 2.9 mm (corresponding to the characteristic impedance *Z*_c_ at the center frequency *f*_c_) is placed on the bottom face of the substrate [15]. Therefore, the ground plane of the microstrip line is on the upper face of the substrate where the butterfly slot is located. The microstrip line is prolonged for the length *l*_me_ over the antenna center to fine-tune the antenna parameters in both antenna versions. The feeding port, with the SMA connector, is set for length *l*_m_ away from the slot center.

In order to obtain an equal input impedance at 2.4 GHz for both antenna ports, dimensions *a*, *a*_1_, and *b* of the slots are optimized as well as the extensions of the CPW *l*_ce_ and microstrip *l*_me_ feeding lines simultaneously examining the impedances at both ports. Figure 3 and Figure 4 display the optimized input impedance of each port separately for the two proposed antennas depicted in Figure 2. It can be noticed that impedance of the slot antenna fed by the microstrip port has three resonances: one with value *Z*_c_ at *f*_c_ and two resonances with very high value. However, the input impedance of the slot antenna fed by the CPW port has two resonances, between which there is impedance near *Z*_c_ around *f*_c_. Also, the optimized resonant frequencies at 2.4 GHz are a little bit different at the CPW and microstrip ports as well as their input impedances.

Figure 5 illustrates the simulated antenna current density on the top side of the substrate when the different ports are excited at *f*_c_. It is obvious that the current flow of the slot antenna is independent of the engaged feeding methods, since it is mainly caused by the slot shape and the mutual position of the two bow-tie slots. It can be also observed that when the microstrip port is excited it simultaneously excites the CPW port but to a lesser extent, which determines the value of the isolation between ports, and vice versa.

Due to the very similar current density of Antenna 1 and Antenna 2 in Figure 5, a simulated electric field distribution only for Antenna 1 is shown in Figure 6. The presented electric field distribution in Figure 6a is obtained for the CPW port excited when the MSL port is terminated by a 50 Ω load. Similarly, Figure 6b presents the electric field distribution for the MSL port excited when the CPW port is terminated by a 50 Ω load.

The change in electric field direction with the change in excitation, especially in the left slot half, is clearly visible in Figure 6. As indicated in Figure 6, vectors of the electric field along the butterfly wings are positioned at the right angle regardless of port excitation: electric field vectors along the left and right wing are at the right angle for CPW port excitation, and electric field vectors along the top and down wing are at the right angle for MSL port excitation.

## 4. Linear Dual-Port Array of the Butterfly Slot Antenna

A high-gain antenna is crucial in some biosensing applications, especially when precision and reliability are important for system functionality. In the case of significant signal attenuation through biological tissue or long-distance directional communication, it is necessary to enhance sensitivity to very weak emitted signals. The antenna array is usually used to achieve the requested high gain. A proposed single-substrate design, without a need for complicated matching circuits, is particularly suitable for a low-cost array design. Unlike rectangular crossed slots where only width and length can be optimized [28,29], the coupled butterfly slot antenna can be simply incorporated in a dual-port array thanks to additional opportunities to adjust antenna resonant frequency and impedance at both ports simultaneously.

As proof of principle, a linear array of the dual-port butterfly slot antennas has been designed in WIPL-D Pro CAD 2023 software [30], and its details are shown in Figure 7.

The array features eight dual-port butterfly slot antennas positioned along the *x*-axis on the top side of the FR4 substrate. The distance between elements in the array is determined mainly by series feeding, in which the input power to the antenna comes from one end of the array by the CPW line. With this end feeding, the main beam angle will be very sensitive to frequency change due to the progressive phase change of the series-fed elements. To avoid the main beam squint at the central frequency, the distance between radiating elements should be equal to the wavelength of the CPW feeding line *d* = *λ*_gcpw_, (*λ*_gcpw_ = 80 mm is the wavelength of CPW feeding line at the center frequency *f*_c_), where *λ*_gcpw_ = 0.64*λ*_0_ and where *λ*_0_ is the free-space wavelength at the frequency *f*_c_ = 2.4 GHz.

Moreover, the butterfly shape of the antenna requires enough space between neighboring slots to prevent overlapping of their wings. The CPW line is extended over the last slot for the length *l*_c._

On the bottom side of the substrate, there is the corporative microstrip network consisting of impedance transformers, widely used in antenna design [31,32], and also enabling uniform distribution for the array’s elements. Its eight microstrip lines extending over the slot center for *l*_m1_ are positioned orthogonally on the CPW line from the substrate top side. The CPW and MS lines have the characteristic impedance *Z*_c_, featuring the dimensions specified in the Section 3.

As proof of concept, an analysis of mutual coupling between two identical butterfly slot antennas positioned at the distance *dis* (center to center) was conducted and is depicted in Figure 8a. The CPW ports are labeled as c while the MSL ports are labeled as m. The minimum distance in the graph in Figure 8 is determined by the length of the butterfly wings preventing their overlap. Also, Figure 8a shows that the influence between MSL ports is greater than the influence between CPW ports or between MSL and CPW ports.

Further analysis studies the mutual coupling between two identical butterfly slot antennas but with one intermediate slot element (Figure 8b). The distance *dis* between antennas is uniform. However, the influence between CPW ports is dominant here.

Figure 9 presents magnitude and phase distribution at the output ports of the MSL corporative feeding network in Figure 7. Figure 10 presents magnitude and phase distribution for the elements in the array obtained by CPW series feeding where, for clarity, only results for the elements of the array with odd ordinal numbers are given. Numerical labels for magnitude and phase distribution match numerical labels of the array’s elements from Figure 7.

Figure 11 shows the optimized input impedance of the array at each port separately. The impedance at both ports has a few resonances in the considered frequency range, but the dimensions of the radiating slots are adjusted to exhibit the impedance *Z*_c_ at frequencies between 2.4 GHz and 2.55 GHz. Figure 12 confirms the results from Figure 11, showing that the simulated *S*_11_ parameter is less than −10 dB for the frequency range 2.4 GHz to 2.54 GHz and the *S*_22_ parameter is less than −10 dB for the frequency range 2.385 GHz to 2.54 GHz. Figure 12 also shows that the simulated *S*_21_ parameter is around −20 dB in the range between 2.4 GHz and 2.54 GHz.

Figure 13 and Figure 14 depict simulated radiation patterns in the *xz*-plane (*φ* = 0°) and in the *yz*-plane (*φ* = 90°) at the frequency of 2.4 GHz with feeding by the CPW and by microstrip port, respectively. The array fed from a CPW port has maximum gain 14.6 dBi and the SLS (side lobe suppression) of 11.9 dB in the E plane. As an opposite to the single dual-port slot antenna, the antenna array fed from an MSL port has a greater maximum gain of 16.7 dBi and the SLS of 12.3 dB in the E plane. It is mainly caused by the fact that the MSL port feeds the array through a corporative feeding network while CPW is the series feeding type. Although the CPW series feeding offers a simpler feeding network, it requires that the input signal passes through all antenna elements sequentially, which increases overall transmission losses and signal uncertainties.

The most important radiation characteristics of the linear dual-port array are given in Table 1. Keeping in mind all presented results, the linear dual-port array can be considered as a good choice for many full-duplex biosensing applications requesting a dual-port antenna as a sink node that is required to feature high performances, such as high gain [33].

## 5. Experimental Results of a Single Antenna

The proposed two dual-port butterfly slot antennas shown in Figure 2 are simulated and optimized using WIPL-D Pro CAD software [30] to obtain the antenna’s input impedance equal to the characteristic impedance of the feeding lines *Z*_c_. However, the reflector plate, which has the same dimensions as the substrate, does not achieve an adequate front-to-back ratio. As a result, the size of the reflector plate is optimized to enhance the antenna’s front-to-back ratio.

The magnitudes of the simulated *S*_11_ and *S*_22_ parameters, corresponding to feeding by the CPW port and microstrip port, respectively, are displayed in Figure 15.

The simulated *S*_11_ is less than −10 dB for frequency range 2.34 GHz to 2.525 GHz for Antenna 1 and 2.23 GHz to 2.415 GHz for Antenna 2. Similarly, the *S*_22_ is less than −10 dB for frequency range 2.34 GHz to 2.51 GHz for Antenna 1 and 2.275 GHz to 2.475 GHz for Antenna 2. It can be seen that all results in Figure 15 correspond very well with the simulated results for the input impedance shown in Figure 3 and Figure 4. Since both antennas have very similar simulated *S*_21_ parameters, as shown in Figure 16, only Antenna 1 is fabricated and measured.

Photographs of the fabricated antenna prototype are depicted in Figure 17. Measurements of the antenna prototype, conducted in a wide frequency range, demonstrate a strong correlation with the simulation results. The magnitudes of the measured *S*_11_ and *S*_22_ parameters are depicted in Figure 15. The measured impedance bandwidth (*S*_11_, *S*_22_ < −10 dB) is from 2.35 GHz to 2.55 GHz, which is 8.33% of the center frequency *f*_c_. The measured isolation between ports (magnitude of *S*_21_ parameter) is greater than 25 dB for the frequency range between 2 GHz and 2.47 GHz (Figure 16). In general, S-parameters obtained by antenna measurement show very good agreement with simulations shown in Figure 15 and Figure 16. A small frequency shift between simulation and measurement of about 25 MHz for *S*_11_, 40 MHz for *S*_22_, and 10 MHz for the *S*_21_ parameters is likely due to printing tolerances and cable and connector effects during the measurements.

The measured peak gains *G* and efficiency for the CPW and MS ports are presented in Figure 18 and Figure 19, respectively. The measured peak gains *G* is obtained using the measured *S*-parameters of two identical fabricated antennas (*G* = *G*_1_ = *G*_2_) at the distance *r* = 1.76 m by the formula:(1)$${\left(\frac{\lambda_{0}}{4\pi r}\right)}^{2}G_{1}G_{2}=\frac{{\left|S_{21}\right|}^{2}}{\left(1-{\left|S_{11}\right|}^{2}\right)\left(1-{\left|S_{22}\right|}^{2}\right)}$$

The value of peak gain at the CPW feeding port varies between 7 dBi and 8.5 dBi within the impedance bandwidth. However, peak gain is a bit lower when the microstrip port feeds the antenna, and it fluctuates between 6.5 dBi and 7.75 dBi in the same bandwidth. The greater peak gain, when the CPW port is employed, is also obtained by simulation (Figure 18 and Figure 19). Therefore, it can be concluded that the obtained measured peak gains are in close agreement with the simulated values. Also, for both ports, CPW and MSL, measured efficiency is greater than 80% in the whole operating band, with a value of 90% for the CPW port and 83% for the MS port at *f*_c_.

The measured radiation patterns in the *xz*-plane (*φ* = 0°) and in the *yz*-plane (*φ* = 90°) with feeding by the CPW port and microstrip port for the frequencies 2.3 GHz, 2.4 GHz, and 2.5 GHz are depicted in Figure 20 and Figure 21, respectively. Although there are slight differences in the backside radiation that can come from the measuring system and environment, which is generally at a low level, the simulated and measured patterns show good agreement for both feeding ports used. Moreover, the radiation patterns in examined planes, the *xz*-plane (*φ* = 0°) and *yz*-plane (*φ* = 90°) for the CPW port, are almost the same as the *yz*-plane and *xz*-plane radiation patterns for the microstrip port. Therefore, the E and H planes’ radiation patterns for the CPW port are almost the same as the radiation patterns in the H and E planes for the microstrip port. Also, both port excitations feature similar radiation patterns in the E and H planes. In summary, it is observed that the proposed antenna can simultaneously receive both vertical and horizontal polarization at different antenna ports.

Table 2 gives the review of the most important measured characteristics of the proposed antenna. It can be noticed that measured port isolation is around 25 dB and the measured efficiency is above 80% in the whole impedance bandwidth. Also, the gain fluctuates between 6.5 dBi and 8.5 dBi in the concerned frequency range. But, the front-to-back ratio is mostly between 15 dB and 20 dB, except for a few measurements at 2.3 GHz. These measured results are important if the antenna is intended for application in the systems sensitive to unwanted electromagnetic radiation. Also, if the antenna is included in WBAN wireless system, it will have less effect on the body, due to its reduced back-feeding radiation. Further, cross-polarization level is lower for the CPW port, and it ranges from −23 dB to −13.5 dB. If the antenna is fed from the microstrip port, the cross-polarization level is between −20 dB and −10 dB.

Moreover, the antenna rotated for ±45 degrees can be used for precise polarimetric Doppler radar respiratory measurements [9] to enhance the signal to noise ratio (SNR) of the systems. Figure 22 shows V-polar and H-polar radiation patterns for the antenna rotated for 45 degrees.

It can be perceived that signals with both polarizations appear equally at both antenna ports, which differ mutually in broadside direction.

Table 3 compares the proposed antenna and six previously published dual-port antennas mostly designed at the similar frequency range. All considered antennas are chosen to meet the specific requirements set by full-duplex in-band applications in Doppler biosensing radars: dual-polarized dual port, with reflector to suppress back radiation, and with identical radiation patterns in the E and H planes, i.e., identical corresponding radiation patterns at both ports. The antennas with demanding feeding structure design require two or more substrates and more conductive layers [16,17,18,19,20], which results in complex assembly and increases modeling and manufacturing time and overall antenna cost. The antenna presented in [21] is realized on one substrate with two conductive layers but with large overall dimensions. Additionally, it is fed by two coax lines whose outer conductor is joined to one antenna point while the inner conductor is connected to another antenna point, demanding short pins to enable a 180-phase difference between the inner and outer conductors’ excitations. In contrast, the proposed antenna with only one substrate and two conductive layers uses an extremely simple, in-plane feeding structure consisting of a very short microstrip and CPW line and achieves a comparable performance. Unlike some antennas from Table 3 that are designed to feature broadband characteristics, the proposed antenna has a narrow bandwidth because it was designed for applications at a 2.4 GHz ISM band. However, the antenna is easily scalable for other frequency bands [14,15]. The array of proposed antennas fed by the microstrip line enabling tapered distribution was presented in [34] and the array with a CPW corporative feeding network with uniform distribution was shown [35], both intended for a frequency range from 24 GHz to 27 GHz.

The use of a single substrate with two conductive layers makes this design easy to fabricate. There is no need for additional design and mounting of a supplementary feeding network and components like balun [17], interdigital structures [18], power divider [20], etc., which introduce attenuation, increase antenna size, and make the fabrication more expensive.

## 6. Conclusions

A novel, single-substrate, dual-port, and dual-polarized printed antenna with butterfly radiating slots is presented. By employing only short, mutually orthogonal CPW and microstrip feed lines at the different ports, the proposed antenna is suitable for full-duplex in-band applications in Doppler biosensing radars. Unlike a previously published dual-port crossed-slot antenna where only the length and width of the slots can be adjusted during optimization [17,24], the proposed slot antenna consists of two coupled butterfly arms that allow effective and quick modeling with seven degrees of freedom. In addition to the dimensions of slots that are defined by length and two widths, the parameters for the coupling region and for the extensions of the CPW and MS lines provide more opportunities to adjust the antenna resonant frequency and impedance at both ports simultaneously. This leads to a simple feed structure on the same substrate as the radiating slots, without the additional matching circuits that are needed in other two-port antennas, making the design more compact and suitable for array configurations. This feature was demonstrated through the design of a dual-port eight-element linear array. Similar to the proposed single antenna, the linear array is also a dual-port antenna suitable for full-duplex in-band biosensing applications.

The experimental results show that the single dual-port antenna exhibits very similar radiation patterns for both the horizontal and vertical polarizations. The proposed antenna is scalable due to its very simple design and can be easily redesigned for other frequency bands.

## Figures and Tables

**Figure 1 sensors-25-04980-f001:**
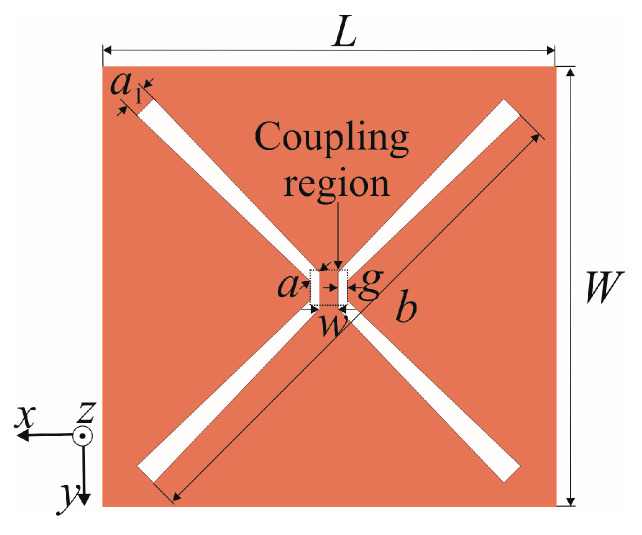
Butterfly slot radiating element with dimensions *a* = 2.06 mm, *a*_1_ = 2.85 mm, *b* = 87.55 mm, *L* = 70 mm, *W* = 66 mm.

**Figure 2 sensors-25-04980-f002:**
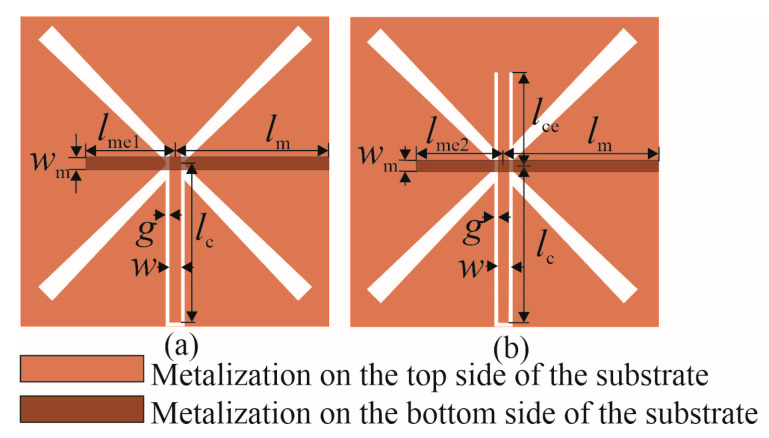
Double-port butterfly slot antennas: (**a**) Antenna 1 and (**b**) Antenna 2.

**Figure 3 sensors-25-04980-f003:**
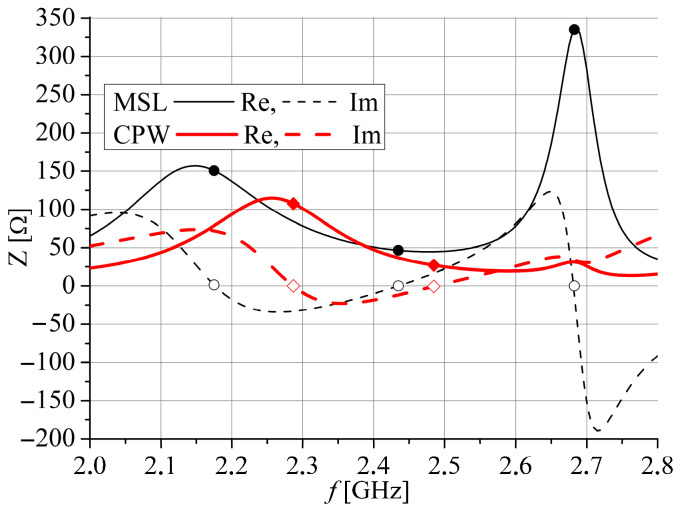
The optimized input impedance of Antenna 1. Resonant frequencies are marked by cycles.

**Figure 4 sensors-25-04980-f004:**
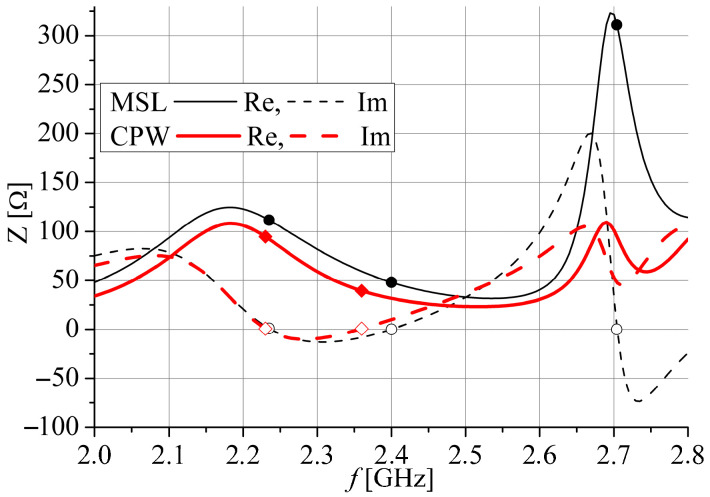
The optimized input impedance of Antenna 2. Resonant frequencies are marked by cycles.

**Figure 5 sensors-25-04980-f005:**
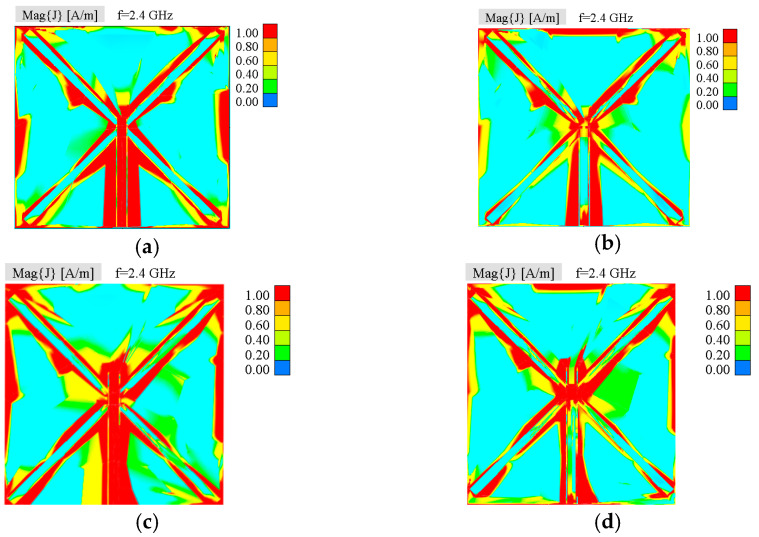
Simulated current density on the antenna upper surface for two proposed antennas: Antenna 1 (**a**) CPW port excited, (**b**) microstrip port excited; Antenna 2 (**c**) CPW port excited (**d**) microstrip port excited.

**Figure 6 sensors-25-04980-f006:**
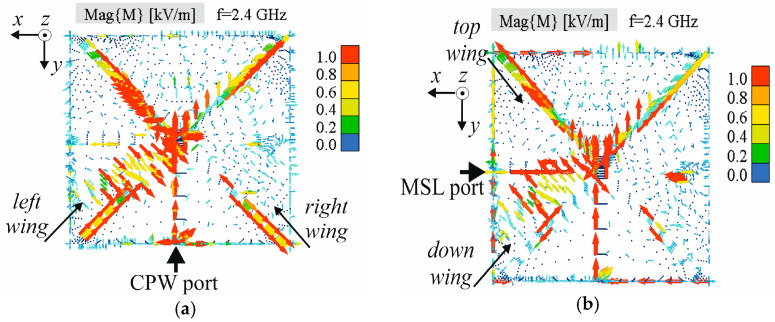
Simulated electric field distribution on the antenna upper surface: (**a**) CPW port is excited, (**b**) MSL port is excited.

**Figure 7 sensors-25-04980-f007:**
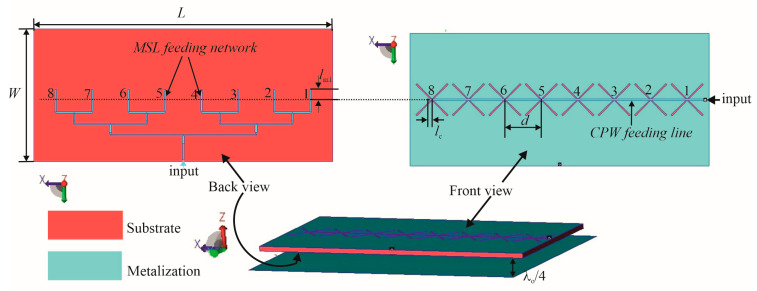
Linear dual-port array of the butterfly slot antenna with dimensions *a* = 2.3 mm, *a*_1_ = 3.8 mm, *b* = 97.4 mm, *L* = 656 mm, *W* = 290 mm, *l*_m_ = 13 mm, *l*_c_ = 8 mm.

**Figure 8 sensors-25-04980-f008:**
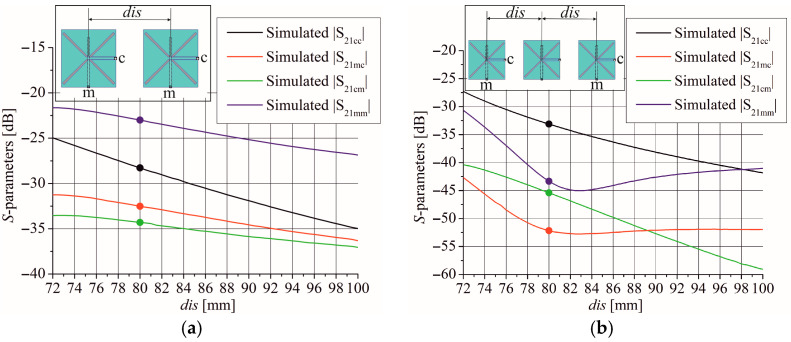
Mutual coupling between (**a**) two identical butterfly slot antennas and (**b**) two identical butterfly slot antennas with one intermediate slot element. Circles denote the mutual coupling at the distance 80 mm that is used in array design.

**Figure 9 sensors-25-04980-f009:**
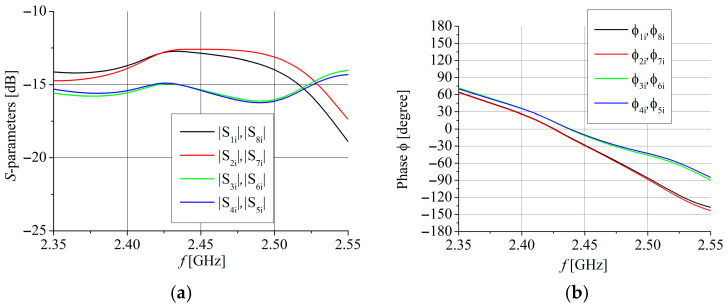
Magnitude and phase distribution for array’s elements for MSL corporative network. (**a**) Magnitude distribution (**b**) Phase distribution.

**Figure 10 sensors-25-04980-f010:**
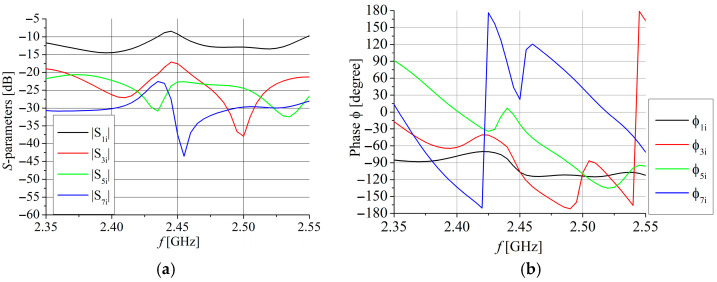
Magnitude and phase distribution for array’s elements for CPW series feeding. (**a**) Magnitude distribution (**b**) Phase distribution.

**Figure 11 sensors-25-04980-f011:**
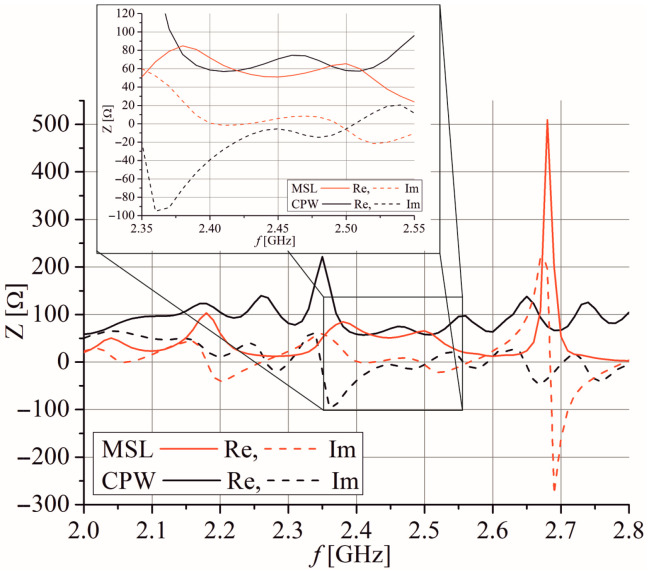
The optimized input impedance of the linear dual-port array.

**Figure 12 sensors-25-04980-f012:**
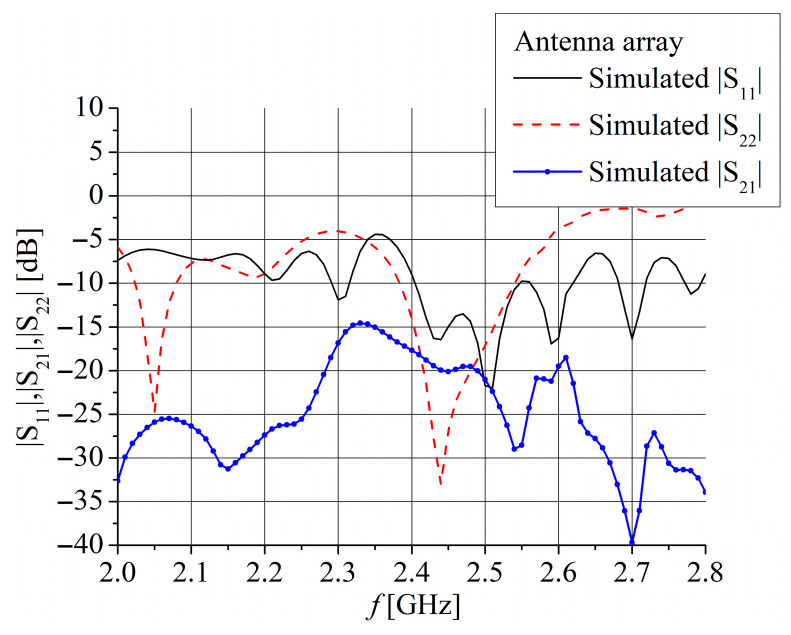
Simulated *S*_11,_
*S*_21_, and *S*_22_ parameters for the linear dual-port array. Port 1 is CPW port and Port 2 is MS port.

**Figure 13 sensors-25-04980-f013:**
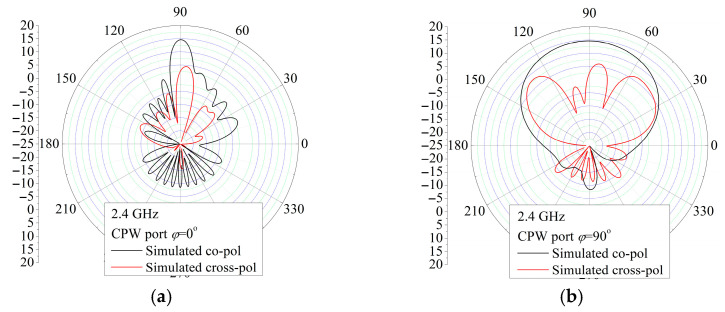
Simulated radiation pattern of the linear dual-port array for CPW port. (**a**) E (*φ* = 0°) plane (**b**) H (*φ* = 90°) plane.

**Figure 14 sensors-25-04980-f014:**
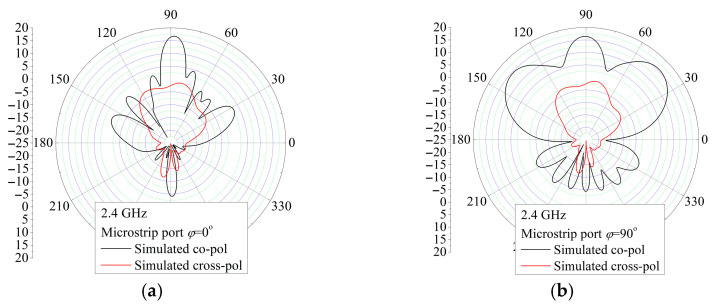
Simulated radiation pattern of the linear dual-port array for MSL port. (**a**) E (*φ* = 0°) plane (**b**) H (*φ* = 90°) plane.

**Figure 15 sensors-25-04980-f015:**
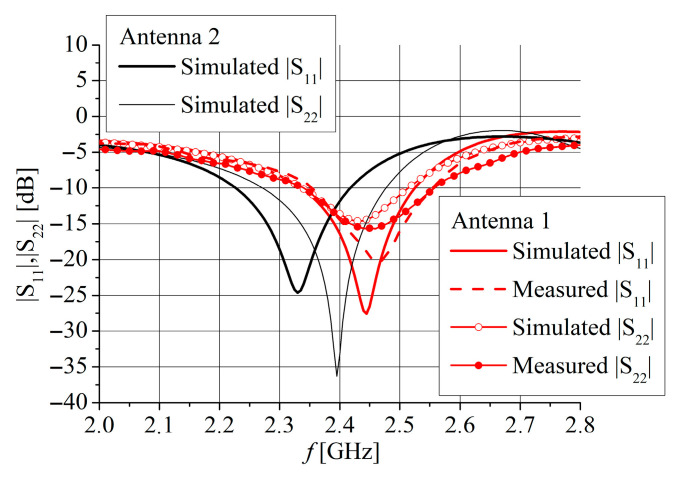
Simulated and measured *S*_11_ and *S*_22_ parameters for Antenna 1 (red color) and Antenna 2 (black color). Port 1 is CPW port and Port 2 is MS port.

**Figure 16 sensors-25-04980-f016:**
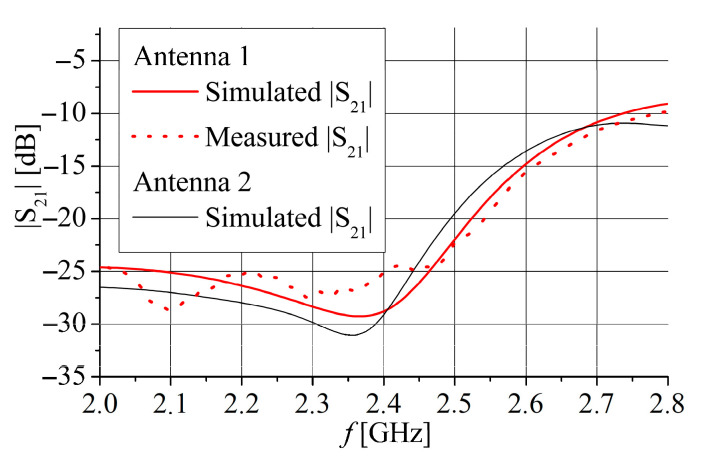
Simulated and measured *S*_21_ parameter for Antenna 1 (red color) and Antenna 2 (black color).

**Figure 17 sensors-25-04980-f017:**
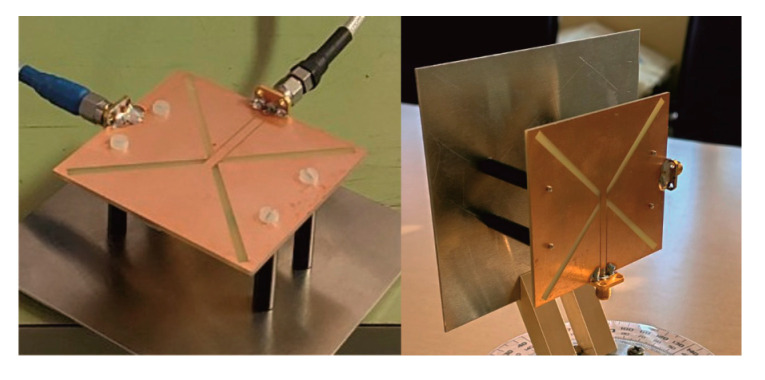
Photographs of the fabricated antenna 1.

**Figure 18 sensors-25-04980-f018:**
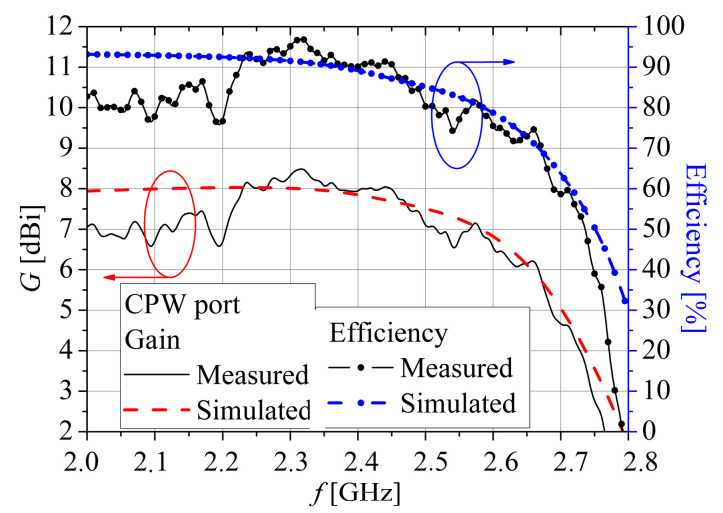
Simulated and measured gain and efficiency for the CPW port.

**Figure 19 sensors-25-04980-f019:**
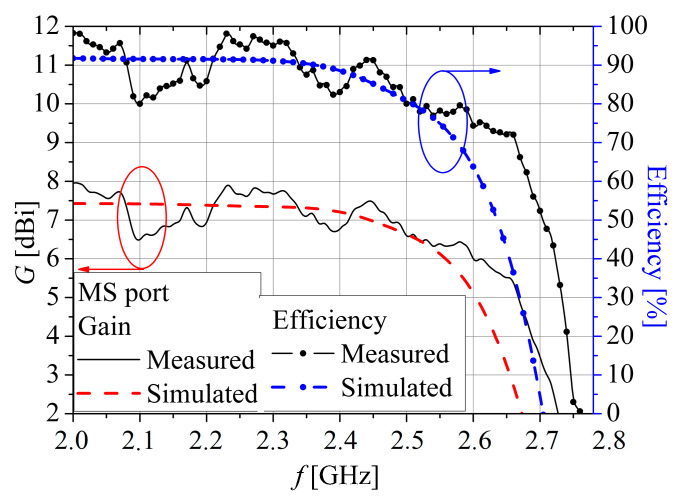
Simulated and measured gain and efficiency for the MS port.

**Figure 20 sensors-25-04980-f020:**
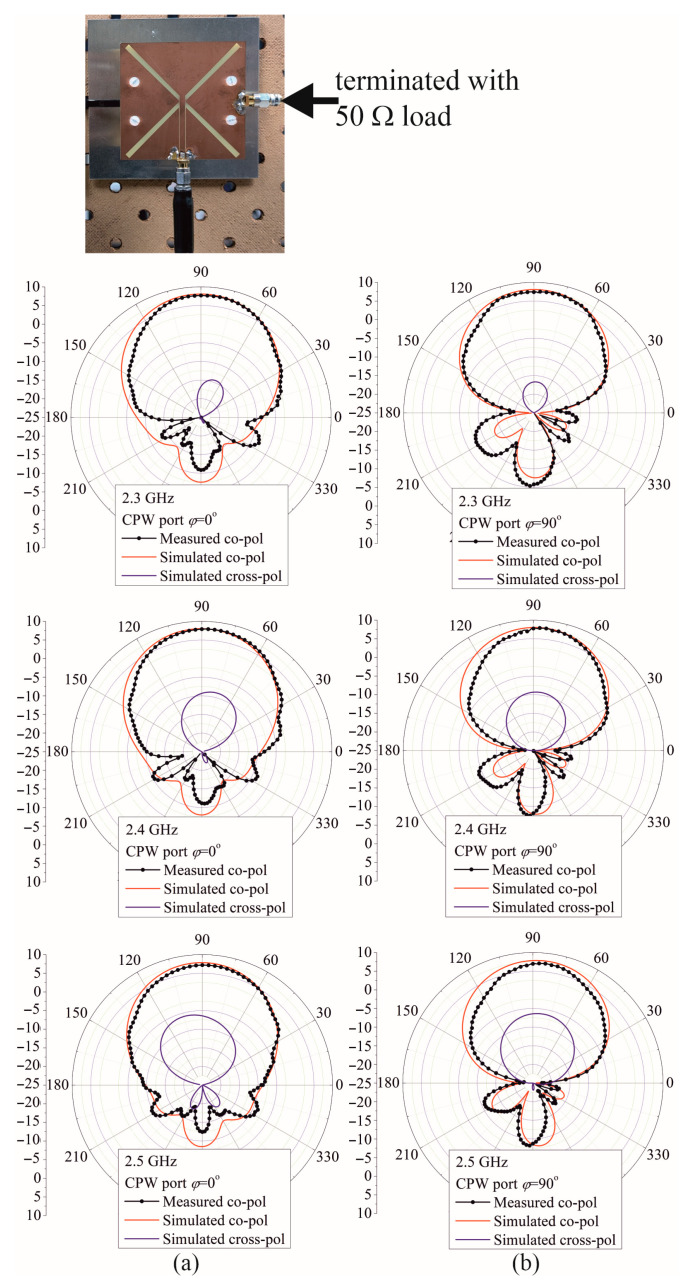
Simulated and measured radiation pattern for the CPW port vertical. (**a**) H (*φ* = 0°) plane (**b**) E (*φ* =90°) plane.

**Figure 21 sensors-25-04980-f021:**
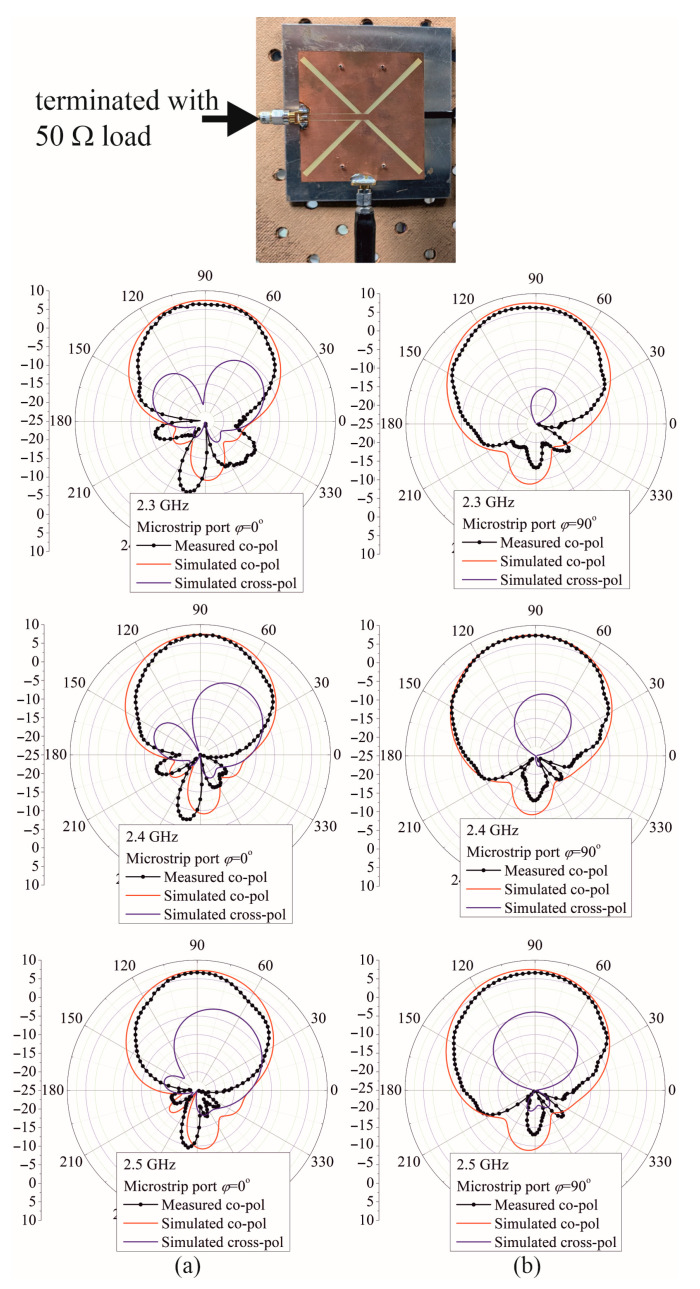
Simulated and measured radiation pattern for the MSL port vertical. (**a**) E (*φ* = 0°) plane (**b**) H (*φ* = 90°) plane.

**Figure 22 sensors-25-04980-f022:**
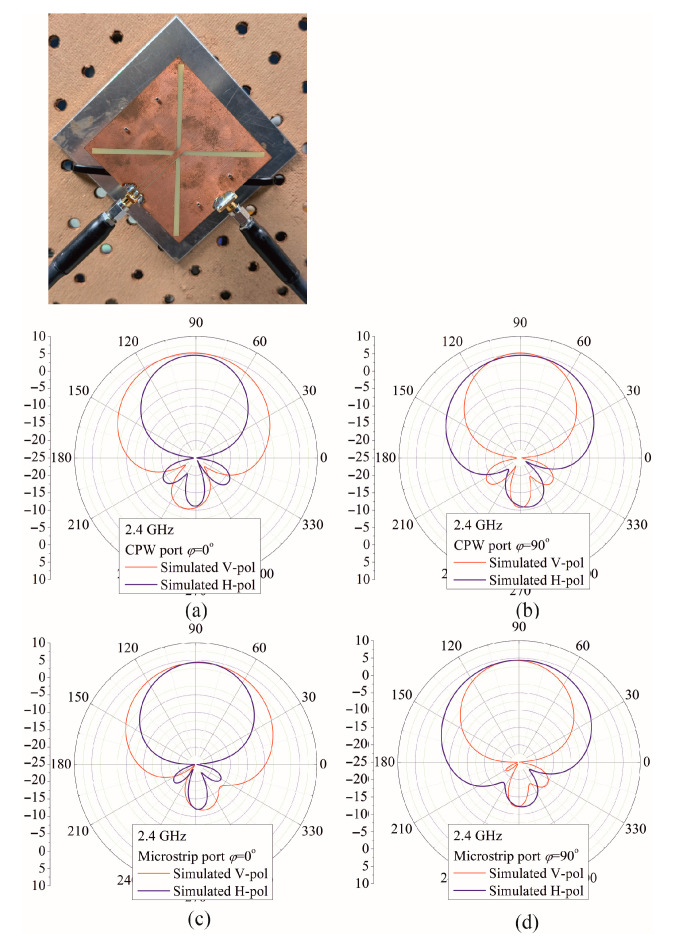
Simulated radiation pattern of the antenna rotated for ±45 degrees for the CPW port: (**a**) H (*φ* = 0°) plane (**b**) E (*φ* = 90°) plane; for MSL port: (**c**) E (*φ* = 0°) plane (**d**) H (*φ* = 90°) plane.

**Table 1 sensors-25-04980-t001:** Characteristics of the proposed array.

Port Isolation [dB]	18
CPW feeding	
Peak gain [dBi]	14.6
SLS in for *φ* = 0° [dB]	11.9
3dB-beamwidth for *φ* = 0° [°]	10
Front-to-Back Ratio for *φ* = 0° [dB]	23
Front-to-Back Ratio for *φ* = 90° [dB]	22.9
Cross-polarization [dB] for *φ* = 0° [dB]	−10.1
MSL feeding	
Peak gain [dBi]	16.7
SLS in for *φ* = 0° [dB]	12.3
3dB-beamwidth for *φ* = 0° [°]	10.6
Front-to-Back Ratio for *φ* = 0° [dB]	21
Front-to-Back Ratio for *φ* = 90° [dB]	20.8
Cross-polarization [dB] for *φ* = 0° [dB]	−15.6

**Table 2 sensors-25-04980-t002:** Measured characteristics of the proposed antenna.

Frequency [GHz]	2.3	2.4	2.5
Port Isolation [dB]	27	25	22.5
CPW feeding			
Peak gain [dBi]	8.5	8	7.1
Efficiency [%]	95	90	80
Front-to-Back Ratio for *φ* = 0° [dB]	18.5	19.1	19.6
Front-to-Back Ratio for *φ* = 90° [dB]	13.3	15.9	15.4
Cross-polarization [dB]	−23	−17	−13.5
Microstrip feeding			
Peak gain [dBi]	7.75	6.75	6.5
Efficiency [%]	95	83	80
Front-to-Back Ratio for *φ* = 0° [dB]	12	14.5	16.1
Front-to-Back Ratio for *φ* = 90° [dB]	19.5	20.3	19.7
Cross-polarization [dB]	−20	−15	−10

**Table 3 sensors-25-04980-t003:** Comparison of the presented results with previously published results.

References	[16]	[17]	[18]	[19]	[20]	[21]	This Work *
Center Frequency [GHz]	2.2	2.2	5.8	2.2/4.25	2.2	2.7	2.4
Number of ports	2	2	2	2	2	2	2
Gain [dBi]	8.3–10.8	7.5	5.8	9.3/9.8	9.3	>9.3	6.75/8
Port Isolation [dB]	>30	20–35	>45	21–30/21–40	>48	>38	25–28
Fractional Bandwidth [%]	55.4	46.3	3.1	46.8/49.4	48.3	46.8	8.3/8.3
FBR [dB]	>15	>11	16	>15	>18	>19	>15
Efficiency [%]	>90	>89	96	-	>86	87	83/90
Dimensions in *λ*_0_	0.48 × 0.48	0.73 × 0.73	0.23 × 0.23	0.27 × 0.27	0.62 × 0.62	1.43 × 1.43	0.85 × 0.8
Profile in *λ*_0_	0.13	0.28	0.07	0.18	0.23	0.25	0.25
No. of substrates	2	3	2	2	3	1	1
No. of conductive layers	3	4	3	4	6	2	2

* Measured results at two antenna ports.

## Data Availability

Data are contained within the article.

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
