# Peer review of "Dual-Port Butterfly Slot Antenna for Biosensing Applications"

_sensors, 2025, doi:10.3390/s25164980_

Round 1
Reviewer 1 Report
Comments and Suggestions for Authors
The authors claim that the main contribution of their work lies in the use of a single-substrate, dual-port butterfly slot antenna. However, similar designs already exist, employing dual-polarized, dual-port, compact antennas. They state that their proposed design is “the simplest proposed dual-port, dual-polarization antenna designed on a single substrate with two conductive layers.” However, this assertion is not supported by a clear discussion of performance trade-offs. As it stands, the reviewer is unable to discern the actual contribution of the work from its current presentation.
Moreover, the extension from a single antenna to an array is straightforward and lacks innovation in terms of the feeding mechanism, phase control, or beamforming techniques. I recommend including a Related Works section that reviews recent advancements in the literature to better position the authors’ contribution. The Introduction should also explicitly state how this work compares to and improves upon existing approaches.
Given the dual-port design, I found the absence of a mutual coupling analysis for the array configuration to be a significant omission.
While the authors propose extending the antenna into an array configuration, no prototype is presented, nor is this approach mentioned in the paper’s title. I suggest reconsidering whether the array extension should be part of this work.
The experimental results are presented but only briefly discussed. The discrepancies between the simulated and measured results are not addressed. I recommend revising this section to provide a more in-depth analysis. Additionally, the paper lacks a study on human body proximity effects, which is particularly relevant given the biosensing application context.
The comparison table is biased in favor of the proposed antenna. Additional metrics such as fractional bandwidth, scanning performance, and beam-steering capabilities should be included to offer a more balanced evaluation.
Finally, I urge the authors to avoid subjective phrasing. A technical report should present results objectively and demonstrate the proposed antenna’s performance through rigorous comparison with existing works to highlight its contribution.
Author Response
Dear reviewer,
We sincerely thank you for reviewing our manuscript entitled "Dual-Port Butterfly Slot Antenna for Biosensing Applications". We are grateful for the constructive comments and insightful suggestions which have been crucial in enhancing the quality of our work.
We have carefully considered all your comments and made corresponding revisions to the manuscript, with the changes clearly highlighted in turquoise. In response, we have provided a detailed, point-by-point reply to each comment, indicating the specific locations of the modifications within the revised manuscript. The primary revisions are incorporated into the manuscript, and our responses to the reviewer’s comments are presented below.
Sincerely,
Marija Milijic
Reviewer: The authors claim that the main contribution of their work lies in the use of a single-substrate, dual-port butterfly slot antenna. However, similar designs already exist, employing dual-polarized, dual-port, compact antennas. They state that their proposed design is “the simplest proposed dual-port, dual-polarization antenna designed on a single substrate with two conductive layers.” However, this assertion is not supported by a clear discussion of performance trade-offs. As it stands, the reviewer is unable to discern the actual contribution of the work from its current presentation.
Answer: The section 2. Related works was introduced where recent advancements in the design of dual-polarized, dual-port, compact antennas are discussed.
Some previous works on dual-port, dual-polarized antenna on one substrate have been added (references [21-24] in the revised paper).
As the aim of the presented work was the 2.4 GHz antenna design intended for comprehensive polarimetric study of physiological Doppler radar signatures where multipath fading problem can occur, it is important that antenna can receive simultaneously both vertically and horizontally polarized waves at the different antenna ports from different direction. Therefore the antenna should have dual-polarized radiation pattern with very similar (in best case identical) radiation patterns in the E and H planes. It can be achieved with two identical radiating elements crossed at the right angle.
Only antenna from reference [21] satisfies requested criteria using one substrate and two conductive layers. However, its feeding methods are implemented through two coax lines each with inner and outer conductors joined the radiator at two points excited in the same strength with a 180 phase difference that made proposed feeding structure more demanding and complicated. Also, its overall dimensions are significantly greater than dimensions of other comparing dual-port dual polarized antenna.
As it was mentioned in previous manuscript version, other considered antennas (references [16-20] in the revised paper) request multilayered structure with more substrates to achieve dual-polarized properties in some frequency range. Their feeding structure are discussed in detailed in Section 2 indicating their complexity of design and fabrication.
Reviewer: Moreover, the extension from a single antenna to an array is straightforward and lacks innovation in terms of the feeding mechanism, phase control, or beamforming techniques. I recommend including a Related Works section that reviews recent advancements in the literature to better position the authors’ contribution. The Introduction should also explicitly state how this work compares to and improves upon existing approaches.
Answer: The section 2. Related works gives the brief discuss about recent work on dual-port antenna array introducing new references [25-27] and leading to the purpose of the proposed array design.
The array presented in [25] needs balun to ensure required elements feeding. There are vias between array elements and corporative feeding network in antenna presented in [26] while 50- coaxial lines between antenna elements and dividers are necessary for antenna design from [27]. Listed components can introduce attenuation, make antenna design more complex and fabrication more expensive.
The array of butterfly slot antenna consists of CPW series and MSL corporative feeding network. The array can be easily realized by simultaneously printing radiating elements and series feeding on top side and corporate feeding network on bottom side.
Moreover, phase control or beamforming techniques can be easily employ for the array featuring simpler feeding methods than the array whose design incorporates balun, vias or coaxial lines.
Reviewer: Given the dual-port design, I found the absence of a mutual coupling analysis for the array configuration to be a significant omission.
Answer: Dependence of mutual coupling between two identical butterfly slots on the distance between their centers is depicted in Fig. 8 in the Section 4. Linear dual-port array of the butterfly slot antenna.
Reviewer: While the authors propose extending the antenna into an array configuration, no prototype is presented, nor is this approach mentioned in the paper’s title. I suggest reconsidering whether the array extension should be part of this work.
Answer: We have carefully considered your suggestion.
Also, the advantage of simple feeding methods of the proposed antenna is additionally demonstrated through its incorporation in the array. As proof of principle, a linear array of the dual-port butterfly slot antennas was proposed considering mutual coupling between elements. Furthermore, based on close agreement between experimental results of a single antenna and simulated results from WIPL-D Pro CAD software, it is expected that experimental results of the array will match closely with presented simulation results.
Reviewer: The experimental results are presented but only briefly discussed. The discrepancies between the simulated and measured results are not addressed. I recommend revising this section to provide a more in-depth analysis. Additionally, the paper lacks a study on human body proximity effects, which is particularly relevant given the biosensing application context.
Answer: The short comments about discrepancies between the simulated and measured results are added in the manuscript in the Section 5. Experimental results of a single antenna. The small differences are noticed for S- parameters which can be due to printing tolerances and cable and connector effect during the measurements. Also, there are slight differences in the backside radiation in the presented radiation patterns that can come from the measuring system and environment.
The antenna and array are not supposed to work near human body. The single antenna is used in in-band full-duplex applications for Doppler radar respiratory measurements (reference [9] presents precise polarimetric measurements of the single antenna use in a full-polarimetric configuration and a dual-receive configuration). The array can be used as a sink node for full duplex biosensing applications. All considered antennas’ employments do not require human body proximity and therefore its effects on the antennas have not been studied. However, Front-to-Back ratio is above 15 dB for the single antenna and above 20 dB for the array indicating their insignificant effect on the body. Greater Front-to-Back ratio is required in this case to protect sensitive radar receiver from antenna back radiation.
Reviewer: The comparison table is biased in favor of the proposed antenna. Additional metrics such as fractional bandwidth, scanning performance, and beam-steering capabilities should be included to offer a more balanced evaluation.
Answer: The bandwidth for all antennas from Table 3 is added. The scanning performances and beam-steering capabilities are not available for antennas from references [16-21] in Table 3. Also, these parameters are not applicable for the presented single antenna.
Reviewer: Finally, I urge the authors to avoid subjective phrasing. A technical report should present results objectively and demonstrate the proposed antenna’s performance through rigorous comparison with existing works to highlight its contribution.
Answer: The section 2. Related works introduces new references with recent advancements in the design of dual-polarized, dual-port, compact antennas. Also, feeding structures of the antennas from Table 3 are discussed in detailed in Section 2 indicating their complexity of design and fabrication. Further, the recent work on dual-port antenna array is introduced through new references [25-27] in section 2.
Reviewer 2 Report
Comments and Suggestions for Authors
This work presents “Dual-Port Butterfly Slot Antenna” for biosensing applications. The paper seems well written in terms of readability, and the addressed topic might not be interesting for the readers of Sensors.
In my opinion, the paper is acceptable considering the following comments:
1. It would be recommended to add electric field distributions for the two polarizations. I think that additional explanation of the polarization of each port is needed.
2. I think the distance between element is too large (1 wavelength) to be used as an array. Have
you considered a method for miniaturization?
3. It would be recommended to add figures of the magnitude and phase of each element for two different feeds with operating frequency band.
Author Response
Dear reviewer,
We sincerely thank you for reviewing our manuscript entitled "Dual-Port Butterfly Slot Antenna for Biosensing Applications". We are grateful for the constructive comments and insightful suggestions which have been crucial in enhancing the quality of our work.
We have carefully considered all your comments and made corresponding revisions to the manuscript, with the changes clearly highlighted in yellow. In response, we have provided a detailed, point-by-point reply to each comment, indicating the specific locations of the modifications within the revised manuscript. The primary revisions are incorporated into the manuscript, and our responses to the reviewer’s comments are presented below.
Sincerely,
Marija Milijic
Reviewer: This work presents “Dual-Port Butterfly Slot Antenna” for biosensing applications. The paper seems well written in terms of readability, and the addressed topic might not be interesting for the readers of Sensors.
In my opinion, the paper is acceptable considering the following comments:
- It would be recommended to add electric field distributions for the two polarizations. I think that additional explanation of the polarization of each port is needed.
Answer: Figure 6. is added to the manuscript with simulated electric field distribution on the antenna upper surface for the proposed antennas Antenna 1 as well as the short discussion of the presented results in the Section 3. Butterfly Slot Antenna.
Reviewer: 2. I think the distance between element is too large (1 wavelength) to be used as an array. Have you considered a method for miniaturization?
Answer: The distance between elements in the array is determined mainly by series feeding in which the input power to the antenna comes from one end of the array by the CPW line. With this end feeding, the main beam angle will be very sensitive to frequency change due to the progressive phase change of the series-fed elements. To avoid the main beam squint at the central frequency the distance between radiating elements should be equal to the wavelength of the CPW feeding line λgcpw=80 mm=0.64λ0 where λ0 is free-space wavelength at the frequency 2.4 GHz.
Moreover, butterfly shape of antenna requests enough space between neighboring slots preventing overlapping of their wings.
Reviewer: 3. It would be recommended to add figures of the magnitude and phase of each element for two different feeds with operating frequency band.
Answer: Figure 9 presents magnitude and phase distribution for elements in the array obtained by MSL corporative network while Figure 10 presents magnitude and phase distribution for elements in the array obtained by CPW series feeding in the Section 4. Linear dual-port array of the butterfly slot antenna.
Reviewer 3 Report
Comments and Suggestions for Authors
This paper presents low-cost, dual-port and dual-polarized slot antennas with high port isolation. Measurement results verifies the design. The comments are listed as below.
Fig.18 the cross polarizations are too high.
Please also add the measurement radiation patterens in Fig.18.
Conclusion part is too long.
Author Response
Dear reviewer,
We sincerely thank you for reviewing our manuscript entitled "Dual-Port Butterfly Slot Antenna for Biosensing Applications". We are grateful for the constructive comments and insightful suggestions which have been crucial in enhancing the quality of our work.
We have carefully considered all your comments and made corresponding revisions to the manuscript, with the changes clearly highlighted in grey. In response, we have provided a detailed, point-by-point reply to each comment, indicating the specific locations of the modifications within the revised manuscript. The primary revisions are incorporated into the manuscript, and our responses to the reviewer’s comments are presented below.
Sincerely,
Marija Milijic
Reviewer: This paper presents low-cost, dual-port and dual-polarized slot antennas with high port isolation. Measurement results verifies the design. The comments are listed as below.
Fig.18 the cross polarizations are too high.
Answer: Fig. 18 (Fig. 22 in the revised paper version) is an example of the antenna usage to enhance the system signal to noise ratio (SNR) for precise polarimetric Doppler radar respiratory measurements that was applied in study presented in reference [9]. The Figure 18 shows V-polar and H-polar radiation patterns when the antenna was rotated for 45 degrees that were mismarked as co-polar and cross-polar radiation patterns. This is corrected in the revised paper version.
It can be perceived that signals with both polarizations (vertical and horizontal) appear equally at both antenna ports.
Reviewer: Please also add the measurement radiation patterens in Fig.18.
Answer: Unfortunately, we do not have access to the laboratory for measurements at the moment of the manuscript revision. But, how it is mentioned in our previous answer, the precise polarimetric Doppler radar respiratory measurements was conducted with the proposed antenna rotated for 45 degrees (as Fig. 22 demonstrates). The obtained experimental results are presented in reference [9] and they prove that signal to noise ratio (SNR) is improved.
Reviewer: Conclusion part is too long
Answer: We omitted some repeated comments and discussion in order to get shorter Section The Conclusion.
Round 2
Reviewer 1 Report
Comments and Suggestions for Authors
I thank the authors for implementing the revision indicated by this reviewer.
Author Response
Dear Reviewer,
Thank you very much for your feedback and thoughtful comments on our manuscript entitled "Dual-Port Butterfly Slot Antenna for Biosensing Applications". We greatly appreciate the time and effort you invested in reviewing the work.
We found your suggestions especially valuable.
Sincerely,
Marija Milijic